# FOOLING THE TEXTUAL FOOLER VIA RANDOMIZING LATENT REPRESENTATIONS

## ABSTRACT

Despite outstanding performance in a variety of Natural Language Processing (NLP) tasks, recent studies have revealed that NLP models are vulnerable to adversarial attacks that slightly perturb the input to cause the models to misbehave. Among these attacks, adversarial word-level perturbations are well-studied and effective attack strategies. Since these attacks work in black-box settings, they do not require access to the model architecture or model parameters and thus can be detrimental to existing NLP applications. To perform an attack, the adversary queries the victim model many times to determine the most important words in an input text and to replace these words with their corresponding synonyms. In this work, we propose a lightweight and attack-agnostic defense whose main goal is to perplex the process of generating an adversarial example in these query-based black-box attacks; that is to fool the textual fooler. This defense, named AdvFooler, works by randomizing the latent representation of the input at inference time. Different from existing defenses, AdvFooler does not necessitate additional computational overhead during training nor does it rely on assumptions about the potential adversarial perturbation set while having a negligible impact on the model's accuracy. Our theoretical and empirical analyses highlight the significance of robustness resulting from confusing the adversary via randomizing the latent space, as well as the impact of randomization on clean accuracy. Finally, we empirically demonstrate near state-of-the-art robustness of AdvFooler against representative adversarial word-level attacks on two benchmark datasets.

## 1 INTRODUCTION

In the last decade, deep neural networks have achieved impressive performance in the Natural Language Processing (NLP) domain. Several deep NLP models have reached state-of-the-art results in several NLP tasks (Devlin et al., 2019) using models such as Recurrent Neural Networks, Transformers, and Pretrained Language Models (PrLMs). However, several works (Iyyer et al., 2018) also reveal that deep NLP models can be fooled by the creation of adversarial examples (Jin et al., 2020). Adversarial examples are synthetically perturbed inputs that are optimized to increase the errors between the predictions of the model and the true labels while being imperceptible to human evaluators (Jin et al., 2020; Iyyer et al., 2018). These developments have sparked concerns about the security and robustness of deep neural networks deployed in NLP applications.

To generate adversarial examples, adversarial attack methods manipulate different aspects of the input sentence, from introducing character errors such as typos or visually similar characters (Gao et al., 2018a; Eger et al., 2019) and replacing words without significantly changing the original semantic in the perturbed inputs (Jin et al., 2020; Li et al., 2020) to recreating sentences with similar meanings using paraphrasing (Iyyer et al., 2018; Qi et al., 2021). Among these methods, adversarial-word substitutions (Alzantot et al., 2018) are some of the most widely studied and effective adversarial approaches. These methods query the victim model multiple times to find the corresponding adversarial perturbations of important words of an input text.

In response to the adversarial threat and to enhance the robustness of NLP models, numerous defenses against textual adversarial attacks have been developed (Zhu et al., 2020; Si et al., 2021; Wang et al., 2020; Ye et al., 2020; Shi et al., 2020b; Xu et al., 2020). The most popular approaches are adversarial training (Zhu et al., 2020; Dong et al., 2021; Madry et al., 2017), which augments

Table 1: Characteristics of all defenses. $\times$ or $\checkmark$ indicates the method lacks or has the specific characteristic, respectively. AdvFooler satisfies all the characteristics by adding lightweight randomization to the latent space.

| Method | no training required | randomized | trivial inference overhead | no additional network | pluggable |
|---|---|---|---|---|---|
| ASCC | $\times$ | $\times$ | $\checkmark$ | $\checkmark$ | $\times$ |
| InfoBERT | $\times$ | $\times$ | $\checkmark$ | $\checkmark$ | $\times$ |
| FreeLB | $\times$ | $\times$ | $\checkmark$ | $\checkmark$ | $\times$ |
| TMD | $\times$ | $\times$ | $\times$ | $\times$ | $\times$ |
| SAFER | $\times$ | $\checkmark$ | $\times$ | $\checkmark$ | $\times$ |
| RanMASK | $\times$ | $\checkmark$ | $\times$ | $\checkmark$ | $\times$ |
| AdvFooler | $\checkmark$ | $\checkmark$ | $\checkmark$ | $\checkmark$ | $\checkmark$ |

the training data with adversarial examples using an additional optimization step, and randomized smoothing (Ye et al., 2020; Zeng et al., 2021), which replaces the model with its stochastic ensemble based on random perturbations of the input; both of these approaches require training modifications and additional non-trivial training computation (e.g., for adversarial training). Due to the discrete nature of texts, to create adversarial inputs, these defenses also substitute input words with adversarial words sampled from predefined perturbation sets constructed based on the potential attacks instead of gradient-based optimization. This substitution process also aims to preserve the original input semantics. Nonetheless, assuming knowledge of the potential attacks is often unrealistic and impractical.

In this paper, we propose a *lightweight*, *attack-agnostic* defensive method that can increase the robustness of NLP models against textual adversarial attacks. Our defense, called **AdvFooler**, *randomizes the latent representation* of the input *at test time* to *fool the adversary throughout their attack*, which typically involves iteratively sampling of discrete perturbations to generate an adversarial sample. Being a test-time defense with negligible computational overhead, AdvFooler also does not incur any training-time computation. Furthermore, as AdvFooler operates within the latent space of the model, it remains agnostic of the perturbation sets associated with potential attacks. While there are some randomization defenses in NLP (Ye et al., 2020; Zeng et al., 2021), they rely on randomized smoothing with necessary modifications to the model's training to reduce the variance in the model's outputs caused by randomizing the input or embedding space. The advantages of AdvFooler over the other defenses are shown in Table 1. Our contributions can be summarized as follows:

- We propose a lightweight, attack-agnostic, and pluggable defensive method that hinders the attacker's ability to optimize for adversarial perturbations leading to adversarial examples. Consequently, the attack success rate is significantly decreased, making the model more robust against adversarial attacks.

- We provide important theoretical and empirical analyses showing the impact of randomizing latent representations on perplexing the attack process of the adversary.

- We extensively evaluate AdvFooler through experiments on various benchmark datasets and representative attacks. The results demonstrate that AdvFooler is a competitive defense, compared to the existing representative textual adversarial defenses, while being under more constraints, including few modeling assumptions, being pluggable, and incurring negligible additional computational overhead.

## 2 BACKGROUND

### 2.1 ADVERSARIAL ATTACKS IN NLP

Adversarial attacks in NLP can be divided into two categories, depending on what information the attacker has on the victim model. These categories are white-box attacks (Ebrahimi et al., 2018; Cheng et al., 2020) - those that require access to the model's architecture and its parameters - and

black-box attacks (Li et al., 2020; Jin et al., 2020; Li et al., 2018) - those that rely only querying the model for its output. Since the model user rarely shares the underlying model's architecture of their NLP service, let alone their model's parameters, white-box attacks have extremely limited practicality, compared to black-box attacks. In the paper, we will focus on the black-box, query-based attack setting, as it is a more realistic scenario in practice and is commonly studied in similar works (Zeng et al., 2021; Nguyen Minh & Luu, 2022; Zhang et al., 2022). Note that, there are also black-box, transfer-based attacks, which, however, still require knowledge of the model architecture for the attack to be effective (Yuan et al., 2021; Li et al., 2021a).

**Black-box Query-based Attack.** Given the text classification task, a model $f : \mathcal{R}^d \rightarrow \mathcal{R}^{|\mathcal{C}|}$ maps an input $x \in \mathcal{R}^d$ to a logit vector of dimension $|\mathcal{C}|$, where $\mathcal{C}$ is the set of label. The goal of textual adversarial attacks is to search for adversarial examples on which the model makes incorrect predictions. Specifically, given an input sentence $x$, a corresponding adversarial example $x'$ can be crafted to satisfy the following objective:

$$\arg \max_c f(x_{adv}) \neq \arg \max_c f(x)$$
$$\text{s.t.} \quad d(x_{adv} - x) \leq \sigma$$

where $d(x_{adv} - x)$ is the perceptual distance between $x$ and $x_{adv}$, and $\sigma$ is the maximum acceptable distance. The distance function $d$ can measure the number of perturbed words between the original and adversarial input (Gao et al., 2018b). Another popular distance is the semantic dissimilarity between a pair of texts (Iyyer et al., 2018; Li et al., 2020) measured as the cosine distance between the corresponding embedding vectors of the inputs extracted from Universal Sentence Encoder (USE) (Cer et al., 2018).

## 2.2 ADVERSARIAL WORD SUBSTITUTION

Among textual adversarial attacks, adversarial word substitution is one of the most widely studied approaches. These attacks create an adversarial example of an input text $x$ by first identifying the most important words in $x$ and then replacing those words with synonyms from their corresponding synonym sets to cause the victim model to misbehave while aiming to preserve the original semantic meaning. The synonym sets can be created using word embedding models (Mrkšić et al., 2016; Pennington et al., 2014) or using the predictions of a PrLMs given the input sentence (Li et al., 2020). One important feature of these attacks is that both of the aforementioned steps require querying the victim model many times to determine which words are important or which synonyms maximize the model's prediction errors. In the first step, the adversary queries the model to calculate the importance score of each word in $x$ and select the most potential locations (with the highest importance scores) for later perturbations. To find the substituting synonyms that maximize the model's prediction error, the attacks can either perform a greedy search (Li et al., 2020; Jin et al., 2020; Li et al., 2018) or combinatorial optimization algorithm (Zang et al., 2020; Alzantot et al., 2018). Similarly, this step involves a sequence of queries on the victim model. By introducing randomization to the latent representations of queries, our defensive mechanism disrupts the adversary's ability to estimate important words, making their synonym-substituting process significantly harder during the attack.

## 2.3 ADVERSARIAL DEFENSE METHODS

To defend against textual adversarial attacks, several defenses (Wang et al., 2022) have been developed. The goal of an adversarial defense is to improve the robustness of the victim model; that is to achieve good performance on both clean and adversarial examples. Existing adversarial defenses can be divided into two categories: certified (Ye et al., 2020; Zeng et al., 2021; Jia et al., 2019) and empirical (Zhu et al., 2020; Wang et al., 2020; Dong et al., 2021; Li & Qiu, 2020; Zhou et al., 2021; Miyato et al., 2016).

First introduced in Goodfellow et al. (2014), adversarial training is the most often studied empirical defense method. It provides additional regularization to the model and improves the model's robustness when training it with adversarial samples. Several subsequent works study adversarial training for NLP tasks with adversarial examples created on the input space (Ren et al., 2019; Li et al., 2020; Jin et al., 2020). Some adversarial training works improve the model's robustness by introducing adversarial perturbations in the embedding space (Zhu et al., 2020), or by incorporating inductive bias

to prevent the model from learning spurious correlations (Wang et al., 2020; Madry et al., 2017), as well as methods that increase diversity of ensembled models, leading to globally better robustness (Pang et al., 2019). These methods effectively improve the model's robustness without significant compromise to its clean accuracy.

In contrast to the empirical methods, certified defenses can provably guarantee model robustness even under sophisticated attackers. One popular certified-defense approach is randomized smoothing (Ye et al., 2020; Zeng et al., 2021), which constructs a set of stochastic ensembles from the input and leverages their statistical properties to provably guarantee robustness. Differential Privacy (DP) defenses such as Lecuyer et al. (2019) are similar to SAFER and RanMASK, achieving certified robustness through the use of DP framework. Note that RanMASK and DP-defenses incur high training overhead. In contrast, Interval Bound Propagation (IBP) methods (Shi et al., 2020a; Jia et al., 2019; Huang et al., 2019) utilizes axis-aligned bounds to confine adversarial examples. Despite their effectiveness and ability to certify robustness, both approaches rely on access to the synonym sets of potential attacks. Randomized-smoothing methods also modify the training phase to reduce the variance of the outputs of the ensemble. Due to the ensemble, they incur significant computational overhead in the inference phase. In contrast, our defense is lightweight and attack-agnostic, and can better control the variance of the outputs by randomizing the latent representations, instead of via randomization in the input space. A summary of the characteristics of AdvFooler and the existing defenses is shown in Table 1.

## 3 METHODOLOGY

### 3.1 RANDOMIZED LATENT-SPACE DEFENSE AGAINST ADVERSARIAL WORD SUBSTITUTION

Since the adversary relies on querying the victim model multiple times to find adversarial examples, receiving fluctuating feedback from the model will make it significantly harder for the adversary to find the optimal adversarial perturbations. To fool such an adversary, we introduce stochasticity to the model by randomizing the latent representation of an input $x$. Formally, for $h_l$ as the $l-$th layer of the model, we sample an independent noise vector $\epsilon$, which is added to $h_l(x)$ as input to the next layer of the model. Without loss of generality, $\epsilon$ is sampled from a Gaussian distribution $\mathcal{N}(0, \Sigma)$ with $\Sigma$ as a diagonal covariance matrix, or $\mathcal{N}(0, \nu I), \nu \in \mathcal{R}$. The detailed algorithm is presented in Algorithm 1.

Let $f_{\text{AdvFooler}}$ be the proposed randomized model corresponding to the original $f$. When the variance of injected noise is low, we can assume that small noise slightly changes the output logits but does not shift the prediction of the model. In other words, the mean of the randomized model $f_{\text{AdvFooler}}$ with input $x$ is exactly the prediction of $f$ for $x$. Consequently, adversarial samples of $f$ are adversarial samples of $f_{\text{AdvFooler}}$.

### 3.2 EFFECT OF RANDOMIZING LATENT REPRESENTATIONS ON ADVERSARIAL ATTACKS

In textual adversarial attacks, the adversary first determines important words in a sentence $x = \{w_0, \ldots, w_i\}$ using the importance score, which is computed for each word $w_i$ as

$$I_{w_i} = f_y(x) - f_y(x_{/w_i}) \tag{1}$$

where $f_y$ is the logit returned by the model $f$ w.r.t ground-truth label $y$. The adversary then selects words with the highest scores to perturb. Intuitively, this process emulates estimating the gradient of the importance score in the discrete space.

When introducing randomness to the latent space, the important scores are changed. Such changes mislead the attack to select a different set of words to perturb. Perturbing these "less important" words (those that have lower original importance scores before randomization) makes the synonym replacement phase significantly more challenging for the attack, thus reducing the attack's success probability.

**Theorem 1.** *If a random vector $v \sim \mathcal{N}(0, \nu I)$ where $\nu$ is small is added to the hidden layer $h$ of the model $f$ which can be decomposed into $f = g \circ h$, the new important score $I_{w_i}^{new}$ is a random variable that follows Gaussian distribution $\mathcal{N}(I_{w_i}, \nu(\|\nabla_{h(x)} g_y(h(x))\|_2 + \|\nabla_{h(x_{/w_i})} g_y(h(x_{/w_i}))\|_2))$.*

Theorem 1 (proof is in the supplementary material) states that randomly perturbing the hidden presentation leads to a randomized important score. When $\nu$ is high, the randomized importance scores have high variance, which can cause the attack to select the wrong important words.

### 3.3 EFFECT OF RANDOMIZING LATENT REPRESENTATIONS ON CLEAN ACCURACY

If randomization induces substantial fluctuation in the model's output, there is a possibility that the model might predict a wrong label, resulting in a decrease in clean accuracy. Randomized defenses such as SAFER (Ye et al., 2020) and RanMASK (Zeng et al., 2021) introduce randomness to the model in the input space (i.e., randomizing the input tokens). While they are also effective against textual adversarial attacks, the input space of NLP models is discrete. Perturbing discrete inputs can lead to significant changes in the model's outputs (a fact that we will empirically prove in the next section), which results in a different model's prediction. To mitigate this problem, existing randomized input defenses rely on randomized smoothing: train the model with perturbed input and apply ensemble at inference to reduce the variance in the model's output. Nevertheless, these approaches are computationally expensive and require access to the training phase. Conversely, by randomizing the latent representations, AdvFooler induces a significantly smaller variance in the model's output. Furthermore, AdvFooler gives the defender flexibility to select the suitable induced variance without requiring access to the training process. Specifically, given a pretrained classification model and a small clean test set, the defender can select the noise scale $\nu$ at which the variance causes the clean accuracy drops by a chosen percentage (e.g., in our experiments, it is 1%).

---

**Algorithm 1** AdvFooler's Randomized Latent Defense

**Input:** a model $f$, an input sentence $x$, noise magnitude $\nu$.

**Output:** output logit $l$.

1: $z_0 = Emb(x)$
2: **for** layer $l$ in $\{1...L\}$ of model $f$ **do**
3: $\quad \epsilon \sim \mathcal{N}(0, \nu I)$
4: $\quad z_{i+1} = h_l(z_i + \epsilon)$
5: **end for**
6: **return** $z_L$

---

### 3.4 EMPIRICAL ANALYSIS

In this section, we empirically study the effect on the model's performance and attack process of various randomized defenses, including AdvFooler.

**Empirical effect on model's performance.** We compute the differences in cross-entropy losses between those of the randomized model and those of the base model for each sample in the AGNEWS test dataset, which expresses how the predictions vary, under different randomization approaches. Figure 1 shows the loss changes in SAFER, RanMASK, and AdvFooler. As we can observe, in SAFER and RanMASK, perturbing the input tokens induces a higher variance in the loss compared to AdvFooler. Consequently, both of these methods lead to significant drops in the model's performance without adversarial training and ensemble

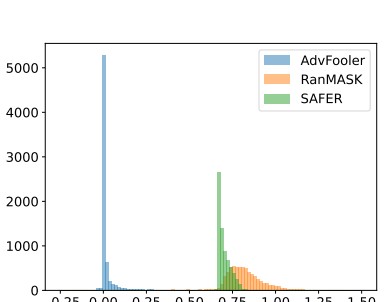
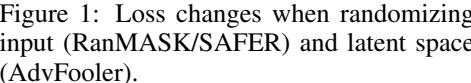
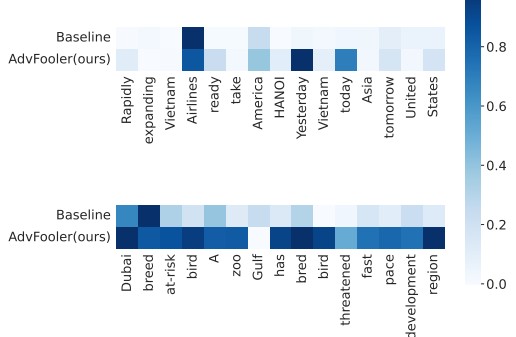

Figure 1: Loss changes when randomizing input (RanMASK/SAFER) and latent space (AdvFooler).

Figure 2: Illustration of each word's important score when calculated with and without Adv-Fooler.

**Empirical effect on fooling the attacks.** To demonstrate that AdvFooler can fool the adversary into selecting non-important words, we calculate the important score of each word for multiple samples from the AGNEWS dataset. The important words before and after applying AdvFooler are presented in Figure 2. As we can observe, AdvFooler causes the attack to select different important words, many of which are originally unimportant. Specifically, in the first sentence, the token "Airlines" is the most important. After adding random noise to the latent representation, AdvFooler changes the important score for each token enough to mislead the adversary into thinking a different

Table 2: The robustness performance of AdvFooler and other defenses on AGNEWS. The best and second-best performances are **bolded** and underlined, respectively.

| Models | Clean% (Drop%) | TextBugger | | | TextFooler | | | BERT-Attack | | |
|---|---|---|---|---|---|---|---|---|---|---|
| | | AuA% (ASR%) | #Query | | AuA% (ASR%) | #Query | | AuA% (ASR%) | #Query | |
| BERT-base (baseline) | 94.20 | 35.20 (62.63) | 351.92 | | 36.80 (60.93) | 317.27 | | 46.50 (50.64) | 337.84 | |
| +ASCC | 90.07 (−4.13) | 39.30 (55.94) | 284.58 | | 41.20 (53.81) | 262.07 | | 45.90 (48.54) | 263.59 | |
| +FreeLB | 95.07 (+0.87) | 47.30 (49.89) | 394.19 | | 49.90 (47.14) | 356.69 | | 52.00 (44.92) | 370.82 | |
| +InfoBERT | 95.01 (+0.81) | 45.20 (52.07) | 373.83 | | 47.50 (49.63) | 336.77 | | **56.00 (40.62)** | 366.40 | |
| +TMD | 94.36 (+0.16) | 47.80 (49.04) | **833.43** | | 51.60 (44.81) | **744.87** | | 52.60 (43.74) | **766.72** | |
| +RanMASK | 90.14 (−4.06) | **52.50 (41.60)** | 582.21 | | **54.60 (38.93)** | 511.98 | | **61.10 (31.73)** | 595.57 | |
| +SAFER | 94.42 (+0.22) | 43.30 (53.74) | 370.77 | | 46.70 (50.05) | 333.17 | | 51.40 (45.03) | 357.47 | |
| +AdvFooler (ours) | 93.67 (−0.53) | 50.10 (45.90) | 819.44 | | 50.10 (45.90) | 701.05 | | 53.40 (42.89) | 752.61 | |
| RoBERTa-base (baseline) | 95.05 | 38.40 (58.93) | 371.20 | | 41.00 (56.15) | 335.88 | | 47.30 (49.41) | 372.99 | |
| +ASCC | 92.61 (−2.44) | 49.50 (46.37) | 372.91 | | 51.00 (44.75) | 330.69 | | 55.70 (39.65) | 374.58 | |
| +FreeLB | 95.03 (−0.02) | 50.50 (46.22) | 448.97 | | 52.00 (44.62) | 402.14 | | 58.20 (38.02) | 442.97 | |
| +InfoBERT | 94.97 (−0.08) | 47.00 (50.11) | 394.56 | | 48.90 (48.09) | 357.44 | | 55.00 (41.61) | 397.35 | |
| +TMD | 93.70 (−1.35) | **52.80 (43.59)** | **878.22** | | **55.20 (41.09)** | **766.74** | | 57.90 (38.14) | **805.19** | |
| +RanMASK | 90.41 (−4.64) | 52.30 (41.50) | 575.94 | | 54.20 (39.37) | 516.98 | | **60.00 (32.43)** | 594.93 | |
| +SAFER | 94.67 (−0.38) | 47.70 (49.15) | 377.67 | | 50.30 (46.43) | 337.81 | | 53.20 (43.22) | 357.56 | |
| +AdvFooler (ours) | 94.21 (−0.84) | 51.70 (44.88) | 804.99 | | 53.30 (42.81) | 717.31 | | 57.80 (38.38) | 767.47 | |

token ("Yesterday" token) is the most important token. A similar phenomenon can be seen in the second sentence, where the important scores change significantly, and unimportant tokens become important ones after randomization. Note that, these experiments use the random noise scale $\nu$ that induces at most 1% clean performance drop.

# 4 EXPERIMENTS

## 4.1 EXPERIMENTAL SETUP

**Evaluation Metrics.** We evaluate the performance of AdvFooler using four evaluation metrics: Clean accuracy (**Clean%**), Accuracy under attack (**AuA%**), Attack success rate (**ASR%**), Average number of queries (**#Query**). Clean% measures the impact of a defense method on the clean performance of the model. Both AuA% and ASR% measure the robustness of the model under attack. #Query measures how many times, on average, the adversary has to query the model to find adversarial examples; the higher #Query is, the more difficult the attack is.

**Datasets.** We evaluate AdvFooler on two benchmark datasets: AG-News Corpus (**AGNEWS**) (Zhang et al., 2015) and Internet Movie Database (**IMDB**) (Maas et al., 2011). AG-NEWS is a classification dataset of news articles created from the AG's corpus[1]. It contains 120,000 training samples and 7,600 test samples where a sample belongs to one of four classes: World, Sports, Business, and Sci/Tech. IMDB is a binary sentiment classification dataset of reviews extracted from the IMDB website. This dataset contains 50,000 samples and is split equally into the training and test sets.

**Model Architecture.** To assess the generality of AdvFooler on model architectures, we evaluate it on two state-of-the-art, pre-trained language models: BERT$_{base}$ (Devlin et al., 2019) and RoBERTa$_{base}$ (Liu et al., 2019). BERT is a Transformer-based language model that outperforms other models in various benchmarks at the time of its release and is widely used for text classification tasks. RoBERTa is a variant of BERT with an improved training process and performance.

**Adversarial Attacks and Parameters.** We choose the following widely used and state-of-the-art adversarial attacks to evaluate our defense: TextFooler (Jin et al., 2020), TextBugger (Li et al., 2018), and BERT-Attack (Li et al., 2020). TextFooler utilizes the word embeddings from (Mrkšić et al., 2016) to generate the set of synonyms for each word, uses both part-of-speech and semantic similarity, and selects words for replacements using important scores. BERT-Attack is similar to TextFooler but uses a pretrained BERT to generate the set of synonyms for a word, given the context of the sentence. TextBugger locates the tokens for perturbations by calculating the important scores for each sentence in the input and then calculating the important score of each token in the sentence. In addition to a synonym set for word substitution, TextBugger also introduces word augmentation - perturbing the characters in words. For a fair evaluation, the adversarial attacks used in the evaluation

---

[1] http://groups.di.unipi.it/~gulli/AG_corpus_of_news_articles.html

Table 3: The robustness performance of AdvFooler and other defenses on IMDB. The best and second-best performances are **bolded** and underlined, respectively.

| Models | Clean% (Drop%) | TextBugger | | TextFooler | | BERT-Attack | |
|---|---|---|---|---|---|---|---|
| | | AuA% (ASR%) | #Query | AuA% (ASR%) | #Query | AuA% (ASR%) | #Query |
| BERT-base (baseline) | 92.14 | 9.20 (90.08) | 500.47 | 11.90 (87.16) | 439.15 | 8.90 (90.40) | 366.52 |
| +ASCC | 88.48 ($-3.66$) | 13.00 (85.68) | 597.07 | 16.90 (81.20) | 529.55 | 7.70 (91.52) | 416.50 |
| +FreeLB | 92.33 ($+0.19$) | 25.80 (71.74) | 776.31 | 28.90 (68.35) | 670.20 | 21.80 (76.12) | 549.85 |
| +InfoBERT | 91.71 ($-0.43$) | 22.50 (75.52) | 719.98 | 25.30 (72.47) | 645.03 | 20.90 (77.26) | 510.46 |
| +TMD | 92.14 ($-0.00$) | 40.40 (56.47) | 3251.52 | 45.10 (51.35) | 2735.58 | 36.20 (60.95) | 2464.83 |
| +RanMASK | 92.61 ($+0.47$) | 31.00 (66.38) | 2740.39 | 35.80 (61.13) | 2392.46 | 33.10 (64.33) | 2463.24 |
| +SAFER | 92.12 ($-0.02$) | **46.10 (50.00)** | 1455.25 | **50.50 (45.64)** | 1262.35 | **43.10 (52.74)** | 1133.61 |
| +AdvFooler (ours) | 91.90 ($-0.24$) | 42.40 (53.41) | 3261.41 | 49.10 (47.32) | 2759.37 | 40.70 (55.76) | 2645.36 |
| RoBERTa-base (baseline) | 93.23 | 6.90 (92.63) | 517.58 | 11.40 (87.79) | 456.35 | 8.20 (91.29) | 439.79 |
| +ASCC | 92.62 ($-0.61$) | 14.70 (84.31) | 770.35 | 20.20 (77.92) | 606.14 | 15.20 (83.57) | 548.34 |
| +FreeLB | 94.20 ($+0.97$) | 25.40 (72.95) | 887.31 | 29.80 (68.30) | 726.84 | 23.10 (75.19) | 647.98 |
| +InfoBERT | 94.18 ($+0.95$) | 20.90 (78.14) | 684.64 | 27.60 (70.61) | 583.97 | 15.50 (83.53) | 518.17 |
| +TMD | 93.22 ($-0.01$) | **66.10 (29.30)** | **4799.56** | **66.40 (28.83)** | 3477.32 | **56.00 (40.36)** | 3330.08 |
| +RanMASK | 94.33 ($+1.10$) | 49.40 (47.45) | 3611.03 | 54.10 (43.29) | 2951.45 | 44.20 (53.67) | 2706.83 |
| +SAFER | 93.75 ($+0.52$) | 62.30 (34.21) | 2059.37 | 63.60 (32.27) | 1874.79 | 54.10 (41.77) | 1415.56 |
| +AdvFooler (ours) | 92.69 ($-0.54$) | 62.20 (33.55) | 4205.77 | 63.70 (32.02) | 3255.35 | 49.40 (47.00) | 3141.88 |

will be based on Li et al. (2021b), which is commonly used by other papers for benchmarking adversarial defenses (Nguyen Minh & Luu, 2022; Zhang et al., 2022)

**Baseline Defenses.** We compare AdvFooler against various types of defense methods. For empirical defenses, we select **Adversarial Sparse Convex Combination** (**ASCC**) (Dong et al., 2021), **InfoBERT** (Wang et al., 2020), **FreeLB** (Zhu et al., 2020), and **Textual Manifold Defense** (**TMD**) (Nguyen Minh & Luu, 2022). ASCC, InfoBERT, and FreeLB improve the model's robustness either by employing adversarial training or introducing additional regularization to the model's training process. ASCC models the word substitution attack space as a convex hull and leverages a regularization term to generate adversarial examples. Different from ASCC, FreeLB introduces adversarial perturbations to word embeddings and minimizes the corresponding adversarial loss around input samples. InfoBERT adds two regularization terms to eliminate noisy information between the input and hidden features and to increase the correlations between the local and global features. Textual Manifold Defense introduces a defense mechanism that trains a mapping function to project the embeddings of input sentences to a set of predefined manifolds to reduce the effects of the adversarial examples. For certified defenses, **RanMASK** (Zeng et al., 2021) and **SAFER** (Ye et al., 2020) are selected. Given an input, both RanMASK and SAFER construct a set of randomly perturbed inputs, using synonym substitution for SAFER and token masking for RanMASK, respectively. Then, they leverage the statistical properties of the predicted output to achieve better model robustness.

**Implementation Details.** We follow the implementation guidelines for evaluating adversarial attacks and defenses in Li et al. (2021b): (1) the maximum percentage of modified words $\rho_{max}$ for AGNEWS and IMDB are 0.3 and 0.1, respectively, (2) the maximum number of candidate replacement words $K_{max}$ is set to 50, (3) the maximum percentage of modified words $\rho_{max}$ for AGNEWS and IMDB must be 0.3 and 0.1, respectively, and (4) the maximum number of queries to the victim model is $Q_{max} = K_{max} * L$, where $L$ is the length of, or the number of tokens in, the input sentence. Clean% is calculated on the entire test set from each dataset, while the robustness metrics AuA%, ASR%, and #Query are computed on 1,000 randomly chosen samples from the test set. All attacks in the experiments are based on the implementation of TextAttack (Morris et al., 2020).

## 4.2 DEFENSE PERFORMANCE AGAINST TEXTUAL ADVERSARIAL ATTACKS

We report the defense performance on AGNEWS in Table 2. As we can observe in this table, on the base model BERT, AdvFooler outperforms all adversarial training techniques (ASCC, FreeLB, and InfoBERT) and the randomized smoothing method, SAFER. AdvFooler is consistently in the top-3 robust defenses; the robustness of AdvFooler and TMD are within 1-2% of each other, while they are generally both lower than the robustness of another randomized-smoothing defense, RanMASK. However, the clean accuracy in RanMASK drops significantly, approximately 4%, while the clean accuracy in AdvFooler is guaranteed to be within 1% of the base model. Note that, all of the evaluated textual defenses require access to the training data. On the base model RoBERTa, AdvFooler similarly achieves a top-3 performance.

For IMDB's results in Table 3, we can observe similar robustness results. TMD's performance is observed to be consistently better with RoBERTa-base. AdvFooler is effective against the attacks and achieves comparable results to the best defenses (SAFER and TMD).

In summary, AdvFooler achieves competitive robustness to the state-of-the-art defenses; however, AdvFooler has significantly lower training and inference overhead compared to TMD, RandMASK, and SAFER (see supplementary material for the computational overhead experiment) and does not access to the training data.

**Difficulty of Adversarial Attacks against AdvFooler.** For all the considered adversarial attacks, the number of queries to locate important words (the first step) of an input text is the same with or without the defenses. Thus, the higher number of queries in an experiment is a result of a more difficult synonym-replacement phase. This also explains considerably smaller average numbers of queries for the baseline (i.e., without any defense) in Tables 2 and 3, compared to the protected models.

We can observe, in Tables 2 and 3, that the average numbers of queries for the adversarial attacks to be successful against AdvFooler are significantly higher than those in all the defenses except TMD in a few cases. The results demonstrate the effectiveness of fooling the attacks in AdvFooler. As explained in Section 3, randomizing the important scores just enough makes the adversary select a different set of important words, which leads to a more challenging synonym-replacement process with a significantly higher number of queries.

**AdvFooler with other attacks.** In the supplementary materials, we provide additional experiments evaluating AdvFooler against other types of attacks, including black-box hard-label (Appendix C.6) and white-box attacks (Appendix C.7), even though they do not align with the threat model studied in this work and several related works (Nguyen Minh & Luu, 2022; Zhang et al., 2022; Zeng et al., 2021; Li et al., 2023) to AdvFooler. The results show that AdvFooler is still effective in defending against these attacks, further highlighting the versatility of AdvFooler.

## 4.3 ROBUSTNESS FROM RANDOMIZING DIFFERENT LATENT SPACES

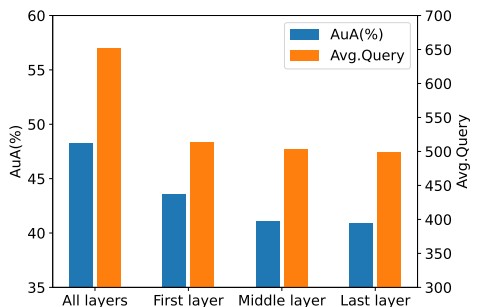

Figure 3: The robustness of AdvFooler when randomizing different layers of the model on AG-NEWS.

In this section, we study the model's robustness when randomizing different layers of the model. We also decided to study the effects of adding noise to only one layer of the model. For comparison, Specifically, we randomize only the first, middle, and last layers and all layers of the model. The model's robustness against TextFooler is reported in Figure 3. As we can observe from this figure, randomizing the early layer of the model is more effective than randomizing the later layers of the model. More importantly, adding noise to all the layers leads to a significantly higher accuracy under attack and a higher average number of queries.

## 4.4 TRADE-OFF BETWEEN CLEAN ACCURACY AND ROBUSTNESS

As discussed in Section 3, our approach allows the defender to flexibly trade-off between the clean accuracy (or variance in the output logits) and the model's robustness, a feature that is challenging to perform in other defenses. In AdvFooler, this is accomplished via tuning the noise scale $\nu$. A large $\nu$ value leads to both a large variance of the model's output, which can lead to a lower clean accuracy, and a high chance of fooling the adversarial attacks.

We report the model's clean accuracy and robustness when varying the value of $\nu$ in Figure 4. As we can observe, increasing $\nu$ generally increases the model's robustness against all the evaluated attacks while the clean accuracy only slightly changes. However, when $\nu$ reaches a certain value, the clean accuracy begins to decrease, at which point the robustness of the model also decreases. The plot also shows that the noise scale when both accuracy under attack and clean accuracy start to decrease would also be the noise scale that makes the model more robust to all types of attack. This

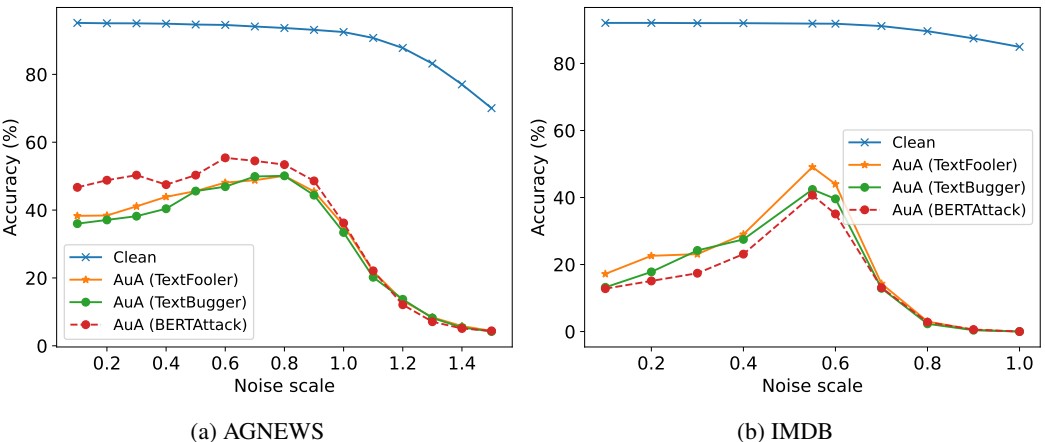

Figure 4: Clean Accuracy and Accuracy under Attack (AuA) when using different noise scales $\nu$.

suggests that, in practice, the defender selects as large $\nu$ value as possible for a certain small drop of clean accuracy. For example, in our experiments, we select the $\nu$ value at which the clean accuracy drops by at most 1% using the test set.

### 4.5 ADVFOOLER WITH ADVERSARIALLY TRAINED MODELS

Table 4: Clean Accuracy and Accuracy under Attack of TextBugger (TB), TextFooler (TF), and BERT-Attack (BA) for FreeLB- and InfoBERT-trained models, with and without using Adv-Fooler .

| Dataset | Models | Clean (%) | AuA(%) | | |
|---|---|---|---|---|---|
| | | | TB | TF | BA |
| AGNEWS | FreeLB | 95.07 | 47.3 | 49.9 | 52.0 |
| | FreeLB+AdvFooler | 94.61 | 62.5 | 63.2 | 63.2 |
| | InfoBERT | 95.01 | 45.2 | 47.5 | 56.0 |
| | InfoBERT+AdvFooler | 94.34 | 64.3 | 64.8 | 67.1 |
| IMDB | FreeLB | 92.33 | 25.8 | 28.9 | 21.8 |
| | FreeLB+AdvFooler | 92.22 | 49.6 | 50.3 | 40.7 |
| | InfoBERT | 91.71 | 22.5 | 25.3 | 20.9 |
| | InfoBERT+AdvFooler | 91.46 | 46.6 | 51.7 | 42.1 |

As discussed previously, AdvFooler can be used to improve the robustness of any existing pretrained model; i.e., AdvFooler is a pluggable defense. In fact, AdvFooler can also be combined with other types of defenses to provide another layer of protection and further improve the model's robustness. In this section, we study the effectiveness of AdvFooler in conjunction with adversarial training methods, including FreeLB and InfoBERT.

Table 4 shows the defense performance before and after applying AdvFooler on adversarially trained models using FreeLB and InfoBERT. We can observe that AdvFooler significantly improves the robustness of the adversarially trained models, with up to 20% improved accuracy under attack in some experiments. While AdvFooler can also be combined with other defensive mechanisms, we leave these to future works.

## 5 CONCLUSION

In this work, we proposed a lightweight, attack-agnostic defense, AdvFooler, that can improve the robustness of NLP models against textual adversarial attacks. Different from existing defenses, AdvFooler does not incur any training overhead nor relies on assumptions of the potential attacks. The main idea of AdvFooler is to mislead word-level perturbation-based adversary into selecting unimportant words by randomizing the latent space of the model, resulting in a significantly more challenging synonym-replacement process. We then provide both theoretical and empirical analyses to explain how AdvFooler fools the adversary, as well as its effect on the model's accuracy. Our extensive experiments validate that AdvFooler improves the robustness of NLP models against multiple state-of-the-art textual adversarial attacks on two datasets. Finally, being a pluggable defense, AdvFooler can also be combined with existing defenses, such as adversarial training, to further protect the NLP models against these threats.

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

# A  PROOF OF SECTION

**Theorem 1.** *If a random vector $v \sim \mathcal{N}(0, \nu I)$ where $\nu$ is small is added to the hidden layer $h$ of the model $f$ which can be decomposed into $f = g \circ h$, the new important score $I_{w_i}^{new}$ is a random variable that follows Gaussian distribution $\mathcal{N}(I_{w_i}, \nu(\|\nabla_{h(x)} g_y(h(x))\|_2 + \|\nabla_{h(x_{/w_i})} g_y(h(x_{/w_i}))\|_2))$.*

*Proof.* When $v$ is small, we have the first-order approximation of the model with noise

$$
\begin{aligned}
f_{\text{AdvFooler}}(x) &= g(h(x) + v) \\
&\approx f(x) + \nabla_{h(x)} g(h(x)) v.
\end{aligned}
$$

In this case, since each application samples a different noise vector, the new important score becomes

$$
\begin{aligned}
I_{w_i}^{\text{new}} &\approx f_y(x) + \nabla_{h(x)} g_y(h(x)) v_1 - f_y(x_{/w_i}) - \nabla_{h(x_{/w_i})} g_y(h(x_{/w_i})) v_2 \\
&= I_{w_i} + v_3
\end{aligned}
$$

where

$$
v_1, v_2 \sim \mathcal{N}(0, \nu I), v_3 \sim \mathcal{N}(0, \nu(\|\nabla_{h(x)} g_y(h(x))\|_2 + \|\nabla_{h(x_{/w_i})} g_y(h(x_{/w_i}))\|_2)).
$$

$\square$

# B  ADDITIONAL IMPLEMENTATION DETAILS

## B.1  DATASET

For IMDB, we use the original dataset [2] from Maas et al. (2011). For AGNEWS, we use the datasets from HuggingFace Datasets (Lhoest et al., 2021). Similar to (Nguyen Minh & Luu, 2022), we also set the model's max sequence lengths for IMDB and AGNEWS to be 256 and 128, respectively based on their average sample length.

## B.2  TRAINING HYPERPARAMETERS

For the NLP models, we employ the pretrained models from HuggingFace Transformer (Wolf et al., 2020) and finetune them for another ten epochs. We use grid search from 1e-5 to 1e-3 to find the optimal learning rate for each model on the respective dataset. The optimally fine-tuned models are used for the robustness evaluation.

# C  ADDITIONAL EXPERIMENT

## C.1  INFERENCE TIME ANALYSIS

In this experiment, we study the computational overhead that each defense introduces at inference time. We use the IMDB test set and record the inference runtime of the BERT model under different defenses. The results are shown in Table 5. From the result, we can see that AdvFooler introduces very little computational overhead compared to most other defenses. Randomized-smoothing methods, such as RanMask and SAFER, incur significant overhead to the inference phase. TMD also adds non-trivial inference overhead (33.3%); as acknowledged by the authors, the main bottleneck of TMD is its high computational overhead of the on-manifold projection (Nguyen Minh & Luu, 2022). Only adversarial training methods, such as ASCC and FreeLB, do not introduce additional overhead during inference time. Most importantly, AdvFooler is the only method that has consistently high robustness and a negligible inference overhead (2.6%).

---

[2] https://ai.stanford.edu/~amaas/data/sentiment/

Table 5: Inference time comparison with other defenses. Tested with BERT on the test set from the IMDB dataset.

| Defense | Runtime (s) | % Increase |
|---|---|---|
| Baseline | 75 | |
| ASCC | 90 | (+20.0%) |
| FreeLB | 75 | (+0.0%) |
| InfoBERT | 75 | (+0.0%) |
| TMD | 100 | (+33.3%) |
| RanMask | 1752 | (+2236.0%) |
| SAFER | 1772 | (+2262.6%) |
| AdvFooler | 77 | (+2.6%) |

## C.2 ADVFOOLER ON VARYING MODEL LAYERS

In this experiment, we study the performance of AdvFooler when randomization is applied at various latent layers of the model. The accuracy and robustness results are shown in Tables 6 and 7. As we can observe from the tables, adding noise to the output of the attention layer yield better robustness in general. Furthermore, injecting noise into the [CLS] token is more effective than to all the hidden layers.

Table 6: Performance results when randomization is applied to different token positions and different locations of the transformer model, on the IMDB dataset. Here, randomization is applied to every self-attention layer.

| PrLMs | Models | Clean% | TextBugger | | BERT-Attack | | TextFooler | |
|---|---|---|---|---|---|---|---|---|
| | | | AuA% (ASR%) | #Query | AuA% (ASR%) | #Query | AuA% (ASR%) | #Query |
| BERT-base | Input embeddings. | 90.78 | 23.40 (74.26) | 2089.57 | 19.50 (78.55) | 1543.31 | 27.80 (69.45) | 1782.04 |
| | Attention output, all tokens | 90.82 | 31.50 (65.12) | 2612.29 | 23.00 (74.59) | 1893.96 | 30.20 (66.33) | 1967.33 |
| | Attention output, [CLS] token | 91.9 | 42.40 (53.41) | 3261.41 | 40.70 (55.76) | 2645.36 | 49.10 (47.32) | 2759.37 |
| | Attention input, all tokens | 91.05 | 24.80 (72.69) | 2259.91 | 18.70 (79.38) | 1707.90 | 28.70 (68.08) | 1886.60 |
| | Attention input, [CLS] token | 91.76 | 22.40 (75.89) | 2153.99 | 21.70 (76.28) | 1759.99 | 25.60 (71.90) | 1781.55 |
| RoBERTa-base | Input embeddings. | 92.02 | 26.80 (70.52) | 2345.55 | 15.90 (82.29) | 1478.49 | 29.80 (66.52) | 2010.82 |
| | Attention output, all tokens | 91.41 | 30.20 (67.17) | 2408.47 | 21.50 (76.61) | 1756.51 | 36.40 (60.56) | 2066.70 |
| | Attention output, [CLS] token | 92.69 | 60.30 (35.58) | 4349.38 | 50.20 (45.55) | 3274.13 | 64.70 (30.20) | 3360.81 |
| | Attention input, all tokens | 92.97 | 23.60 (74.46) | 2086.32 | 15.10 (83.99) | 1426.83 | 28.70 (69.14) | 1747.67 |
| | Attention input, [CLS] token | 92.64 | 27.80 (69.72) | 2504.18 | 18.80 (79.76) | 1553.59 | 33.30 (64.54) | 1927.80 |

Table 7: Performance results when randomization is applied to different token positions and different locations of the transformer model, on AGNEWS dataset. Here, randomization is applied to every self-attention layer.

| PrLMs | Models | Clean% | TextBugger | | TextFooler | | BERT-Attack | |
|---|---|---|---|---|---|---|---|---|
| | | | AuA% (ASR%) | #Query | AuA% (ASR%) | #Query | AuA% (ASR%) | #Query |
| BERT-base | Input embeddings. | 94.57 | 44.80 (52.03) | 711.12 | 45.50 (51.49) | 638.84 | 51.10 (45.23) | 681.39 |
| | Attention output, all tokens | 94.59 | 45.60 (50.97) | 738.90 | 48.60 (48.35) | 652.88 | 54.90 (41.35) | 727.93 |
| | Attention output, [CLS] token | 93.67 | 50.10 (45.90) | 819.44 | 50.10 (45.90) | 701.05 | 53.40 (42.89) | 752.61 |
| | Attention input, all tokens | 94.47 | 44.80 (52.29) | 745.68 | 49.00 (47.71) | 675.86 | 53.10 (43.09) | 731.95 |
| | Attention input, [CLS] token | 93.43 | 34.70 (62.41) | 670.61 | 38.90 (57.90) | 610.20 | 39.40 (57.54) | 640.75 |
| RoBERTa-base | Input embeddings. | 94.72 | 48.90 (47.70) | 752.97 | 48.50 (48.07) | 645.89 | 54.50 (41.90) | 707.38 |
| | Attention output, all tokens | 94.21 | 51.70 (44.88) | 804.99 | 53.30 (42.81) | 717.31 | 57.80 (38.38) | 767.47 |
| | Attention output, [CLS] token | 94.63 | 49.50 (47.17) | 823.36 | 50.20 (46.31) | 718.93 | 59.60 (36.32) | 813.69 |
| | Attention input, all tokens | 93.01 | 40.80 (55.75) | 726.04 | 39.30 (57.00) | 618.76 | 46.10 (50.43) | 672.77 |
| | Attention input, [CLS] token | 94.31 | 40.70 (56.42) | 721.77 | 41.90 (55.14) | 624.96 | 45.60 (50.81) | 681.12 |

## C.3 RANDOMIZATION TECHNIQUES COMPARISON

To show how a lower loss variance makes a randomization method more effective against adversarial attacks, we ran both RanMASK and SAFER, two randomization methods that have a higher variance, through the same experiments on the base model without ensembling. As we can observe from the results in Table 8, both RanMASK and SAFER performed significantly lower than their corresponding baselines in the main experiment. This shows that these methods cannot be used in a similar way to AdvFooler(i.e., without ensembling during inference), since their high variance

Table 8: Performance results for RanMASK and SAFER on the baseline BERT model without ensemble.

| Dataset | Models | Attack methods | | |
|---|---|---|---|---|
| | | TextFooler | TextBugger | BERTAttack |
| AGNEWS | RanMASK. | 0% | 0% | 0% |
| | SAFER | 38.6% | 37.5% | 43.8% |
| IMDB | RanMASK. | 19.5% | 15.6% | 19.3% |
| | SAFER | 28.2% | 21.4% | 18.5% |

makes them more susceptible to attacks, and can only be used in tandem with ensembling and adversarial training with data augmentation.

## C.4 ADVFOOLER AGAINST EXPECTATION OVER TRANSFORMATION (EOT)

Table 9: Accuracy Under Attack of AdvFooler against TextBugger and TextFooler on two datasets using EoT.

| Dataset | Model | EOT results for advfooler | |
|---|---|---|---|
| | | TextBugger | TextFooler |
| AGNEWS | AdvFooler | 50.1% | 50.1% |
| | EOT (same query budget) | 72.2% | 70.2% |
| | EOT (x10 query budget) | 40.5% | 41.0% |
| IMDB | AdvFooler | 42.4% | 40.70% |
| | EOT (same query budget) | 59.5% | 59.7% |
| | EOT (x10 query budget) | 38.6% | 41.2% |

To combat against randomization defense, Athalye et al. (2018) proposed **Expectation over Transformation (EoT)**, a general framework to construct adversarial examples that remain adversarial over a chosen transformation distribution. EoT works by generating an adversarial example from a transformation distribution instead of a single example, which could negate the effects of AdvFooler. However, to capture the distribution of the transformation being made by AdvFooler's noise, the adversary needs to query the model multiple times for each query, thus increasing the query budget. To evaluate AdvFooleragainst EoT-based attacks, we ran our method against the modified TextBugger and TextFooler with the EoT framework. As we can observe in Table 9, the two EoT-based attacks achieve higher Accuracy Under Attack against AdvFooler compared to the original setting. Although EoT can help negate the effects of AdvFooler's randomness on adversarial input, the number of queries needed to find an adversarial example exceeds the budget before it can generate an adversarial example. To re-confirm this explanation, we increased the query budget by a factor of 10. As expected, EoT was able to reduce the Accuracy Under Attack of AdvFooler. Nevertheless, this experiment shows the limited practicality and effectiveness of EoT against AdvFooler.

## C.5 ADVFOOLER'S EFFECT ON MODEL'S OUTPUT ERROR

An immediate question is whether the score returned by AdvFooler is still reliable; otherwise we can just output an arbitrary fixed value for every input, which totally negates any score-based attacks. To calibrate the output of AdvFooler, we calculate the Expected Calibration Error (ECE) of the BERT-base model with and without the use of AdvFooler. We run the experiment five times and report the average ECE score and its standard deviation for comparison.

Table 10: Expected Calibration Errors of the model's prediction, with and without AdvFooler.

| | AdvFooler | Base |
|---|---|---|
| AGNEWS | 2.3±0.2% | 3.9±0.0% |
| IMDB | 6.9±0.2% | 6.3±0.0% |

From the results in Table 10, we can see that, the ECE differences between with and without using AdvFooler is low in both datasets, with the margin of error when using AdvFooler only at 0.2%. This shows that even though AdvFooler can affect the output logits of the model, the changes will only deviate slightly from the original prediction.

## C.6 ADVFOOLER'S PERFORMANCE ON HARD-LABEL ATTACKS.

Table 11: Hard-label results against base models with and without AdvFooler under different noise scale.

|  | Results [AuA%(ASR%)] |
|---|---|
| BERT-base | 58.3% (38.11%) |
| AdvFooler (0.7 scale) | 78.8% (15.35%) |
| AdvFooler (0.9 scale) | 83.3% (9.45%) |
| AdvFooler (1.1 scale) | 85.7% (5.72%) |

In some cases, the adversary can choose to ignore the output logits and only rely on the hard-label output of the model, as seen in Maheshwary et al. (2021).The so-called hard-label attacks first initialize a sample that flips the prediction, move it toward the original input until it passes through the decision boundary, then select the sample at the previous step as adversarial example. However, since AdvFooler is a randomized model, it might sometimes return labels that are different from the model without noise for the attack queries (e.g., instead of yielding class 0, for a hard-label query, the model outputs class 3 because of the random noise addition, especially when the perturbed query is close to the decision boundary of the model). Therefore, it can fool the attack to go a little bit further, and the sample found by the attack turns out to be non-adversarial. In this section, we study the performance of AdvFooler against this hard-label attack. In this experiment, we ran these attacks against the base model and AdvFooler. It is possible that the attack may accidentally flag a chosen perturbation as an adversarial example although this perturbed sample may be predicted incorrectly because of the added noise in our defense; in other words, the selected perturbed sample is not a true adversarial example because, under a different random noise addition, the model prediction may still be correct. To avoid such a situation in our experiment and fairly evaluate the performance of AdvFooler against hard-label attacks, for each "supposedly adversarial example" generated by the attack, we feed it to the model 5 times and record the model's predictions; the model's prediction used for performance evaluation is the prediction occurred most of the times from these 5 applications. As we can observe from Table 11, hard-label attack struggles to find consistent Adversarial Examples the bigger the noise being inserted into the model.

## C.7 ADVFOOLER PERFORMANCES AGAINST THE TRANSFERRED, WHITE-BOX ATTACK

Table 12: Accuracy under Attack and Attack Success Rate of each method against HotFlip.

|  | AGNEWS | IMDB |
|---|---|---|
| BERT-Base | 46.3% (50.85%) | 14.8% (83.83%) |
| FreeLB | 58.9% (37.61%) | 36.8% (60.68%) |
| InfoBERT | 57.5% (39.02%) | 30.0% (67.5%) |
| AdvFooler | 60.2% (34.78%) | 50.7% (44.41%) |

In the main paper, we focus on defending in the black-box setting, as it is a more realistic scenario in practice. Nevertheless, we provide an empirical study of the effectiveness of AdvFooler against a type of attack where the adversary has access to either the base model or the randomized model to generate the adversarial example of an input. We call this type of attack **transferred, white-box** attacks. Note that, this attack is not truly a white-box attack, where the fact that the adversary has access to the base model, but there still exists a mismatch between the attacker's possessed model and the defender's possessed model due to the added randomization. Even with the randomized model, the base white-box approach used in this study is not specifically designed to make good use of the noise distribution of the randomized model, thus, it is not truly a white-box attack case.

Most white-box attacks focus on increasing the loss of the model through gradient computation, choosing the perturbation in the sentence that increases loss the most. We employ **HotFlip** (Ebrahimi et al., 2018) as the studied white-box attack. The experimental setup is similar to that in the main experiments, with the number of words of a sentence that the attacker can perturb increased to 10% and 30% (instead of only two words in the original paper) for IMDB and AGNEWS, respectively. We also give HotFlip access to the model's original parameters, which they can use to create adversarial examples and bypass defense mechanics provided by the defense methods (in the case of RanMASK, TMD, and AdvFooler). To make sure the adversarial examples created using the process can trick the model using the defense, we also ran the adversarial examples through the defense

method 5 times and take the class that being classified to the most. As we can observe in Table 13, AdvFooler is effective against HotFlip in both datasets, increasing the AuA of the baseline model by over 30%. We hypothesize that introducing randomness into the model can turn it into a smoothed classifier, consequently diminishing the adversarial impact. Furthermore, we also consider the case where the adversary ignores the noise from the defense and only attacks the original model. We compute the AuA/ASR of AdvFoolerin this case and compare it to RanMASK and TMD. Table 13 shows that AdvFooler still achieves high robustness compared to TMD.

Table 13: Accuracy under Attack and Attack Success Rate of AdvFooler against HotFlip when ignoring noise.

|  | AGNEWS | IMDB |
|---|---|---|
| AdvFooler | 48.3% (48.50%) | 77.3% (16.61%) |
| RanMASK | 85.3% (3.9%) | 84.1% (9.8%) |
| TMD | 44.2% (52.72%) | 63.3% (29.9%) |

### C.8 COMPARISON TO OTHER DEFENSES WITH SIMILAR ACCURACY

We conduct experiments on tuning the masking rate of RanMASK in AGNEWS such that the clean accuracy is similar to that of AdvFooler. We decrease the random masking rate from 90% of the words in a text example to 30%, which would increase RanMASK's accuracy, as shown in their paper Zeng et al. (2021). We also retrain a RanMASK with a random masking rate of 30%, to prevent differences in masking rate between training and inference.

Table 14 shows that when decreasing the masking rate in RanMASK does increase the clean accuracy of the model, from 90% to 93% in accuracy. On the other hand, the accuracy under attack has decreased tremendously, reducing to half of the original AuA. This shows that, while RanMASK was able to be robust against adversarial attacks, this method also requires larger trade-offs in accuracy compared to other methods. The result also shows that RanMASK requires fine-tuning of the masking rate to achieve good performances under attack, as the effective masking rate was chosen from their certified robustness experiments.

Table 14: The performance of RanMASK on AGNEWS with different mask rates.

| Models | Mask Rate | Clean Accuracy (%) | TextBugger | | | TextFooler | | | BERTAttack | | |
|---|---|---|---|---|---|---|---|---|---|---|---|
| | | | AuA(%) (ASR(%)↓) | Avg. Query↓ | | AuA(%) (ASR(%)↓) | Avg. Query↓ | | AuA(%) (ASR(%)↓) | Avg. Query↓ | |
| BERT | 0.9 | 90.14% | 52.50% (41.60%) | 582.21 | | 54.60% (38.93%) | 511.98 | | 61.10% (31.73%) | 595.57 | |
| BERT | 0.3 | 93.50% | 22.50% (75.88%) | 360.12 | | 24.10% (74.09%) | 325.63 | | 42.90% (54.12%) | 412.99 | |
| RoBERTa | 0.9 | 90.41% | 52.30% (41.50%) | 575.94 | | 54.20% (39.37%) | 516.98 | | 60.00% (32.43%) | 594.93 | |
| RoBERTa | 0.3 | 93.30% | 34.00% (63.56%) | 396.72 | | 35.7% (61.74%) | 359.52 | | 45.60% (50.97%) | 429.72 | |

## D LIMITATIONS

Although AdvFooler is a lightweight, pluggable, and effective defense, there are some limitations that can be improved. Our analysis and empirical evaluation focus on query-based, word-level attacks. We do not include black-box attacks that utilize the transferability from a surrogate to a target model, even though our threat model does not make any assumptions about the network architecture or the training dataset. We also do not consider an adaptive attacker, who makes multiple queries to estimate the importance of each word (i.e., $I_{w_i}$). The expected value of $I_{w_i}$ calculated by this adaptive attacker against AdvFooler (i.e., w.r.t model $f_{AdvFooler}$) will be similar to that of the original importance score of $w_i$ without the defense (i.e, w.r.t model $f$). Evaluating AdvFooler against this adaptive attack in conjunction with existing word-level perturbation-based attacks could be an interesting study. Similarly, besides adversarially-trained approaches, it would also be interesting to study the robustness of the combination of AdvFooler and other types of defenses. Finally, for different model architectures, randomizing different latent layers could yield different effects on the

robustness. This would be interesting to boost the effectiveness of AdvFooler for specific models. We leave these to future works.

## E  ETHICS STATEMENT

The rapid integration of NLP models in various domains and applications has brought significant transformations to our daily lives. Unfortunately, most NLP models face significant vulnerability to textual adversarial attacks, undermining confidence in their deployment and usage. Among the existing textual adversarial attacks, word-level perturbation-based attacks pose a severe threat to the model users since these attacks are the most effective and do not require access to the model architectures or their trained parameters.

To address the risks of these attacks, our work proposes a lightweight, attack-agnostic defense for existing NLP models. Our detailed theoretical and empirical analyses show the defense's comparable performance to the existing state-of-the-art defenses across a wide range of NLP models, textual adversarial attacks, and benchmark datasets. However, our defense does not necessitate any additional computational overhead at both training and inference time and can be used with any existing pretrained models. In summary, the proposed defense can improve the adversarial robustness of existing NLP models against word-level perturbation-based attacks, thereby bolstering user confidence in their utilization for real-world applications.

