# OpenReview forum: "Fooling the Textual Fooler via Randomizing Latent Representations"
_ICLR.cc/2024/Conference — Submitted to ICLR 2024_

### Official Review · Reviewer_2d4f · 2023-10-19

**Soundness:** 4 excellent
**Presentation:** 3 good
**Contribution:** 4 excellent
**Rating:** 8
**Confidence:** 5

**Summary:**

1. GOAL:

* 1A. The problem statement:
It is possible to do black box word level attacks by determining the most important words in the input by repeated query and replacing them (with synonyms).

* 1B. What their solution is:
confuse the adversarial black-box attack with a system called "AdvFooler" that randomizes the latent space.



2. METHOD:
The latent space is randomized by adding an independent gaussian noise vector to the l-th layer of a pretrained classification model. The magnitude of noise is selected based on how much the clean accuracy drops by a chosen percentage on a small held out clean set (1% drop is used in paper). Though it’s not actually stated in the paper, I assume this noise is added during evaluation time (so each adversarial query yields slightly different perturbed outputs, and the attack has a hard time figuring out what words to perturb in it’s attack).

Claims Made:
- does not need more compute during training (True)
- does not rely on assumptions about adversarial perturbation word set for a word substitution attack (True)
- theoretical and empirical results presented on adversarial set and clean accuracy (True, they are presented)

3. RESULTS:
There are many metrics to evaluate on, so we'll go metric by metric: high clean accuracy (A), middling robustness-accuracy tradeoff (B/C), great implementation ease (D), and (probably) low compute time needed which is great (E)

* 3A. By design the clean accuracy does not suffer more than 1% on their method (when used alone), and they show an example where they are still able to change the most significant words.

* 3B. Their method isn't the best method in terms of robustness and accuracy tradeoff, but it is reasonably robust (i.e. does improve robustness significantly). Conclusions based on these comparisons are hard to make to be honest. More details in the Weaknesses section.

* 3C. Some results in Fig 4 demonstrate the tradeoff between clean accuracy and adversarial attack accuracy for this method specifically. Is is unclear why the clean accuracy drops so little at really high noise rates, and that's not really explained.

* 3D. Where their method absolutely shines is how simple the method is to implement and use based on reading the methods section alone. They don't highlight ease of implementation in the paper, but it's absolutely something that would make this method much easier to adopt.

* 3E. They repeatedly highlight low added compute as a big win for this method. They mention they have results about compute usage across the methods in Supplemental Materials, but I was not able to find these. Please add. Still reading the methods section gives a clear picture that the added compute for this method is very negligible. (It's unclear to me if some adversarial training baseline methods are also reasonably negligible in terms of compute time. Computing synonyms is very cheap, and training on some added adversarial examples is also not marginally more expensive in any meaningful way).

**Strengths:**

The primary and significant strength of this paper is that their method is dead simple to implement and use, and takes almost no added compute time. Even if it's not SOTA (and it doesn't actually claim to be SOTA), it is strongly worth disseminating (after adding some missing results: see Weaknesses).

In much the way drop out rates approximate ensemble training and save us all compute time, this method approximates diverse ensembles used to achieve adversarial robustness (ex: the ADP regularizer  https://proceedings.mlr.press/v97/pang19a), and can save some compute time. Because it's so easy to implement and tune for accuracy, it would have high practical impact -- for that reason alone, it's worth disseminating (after some weaknesses are addressed, see below).

**Weaknesses:**

3 weaknesses and 2 baseline suggestions, enumerated below as 1-5. I am very amenable to changing my decision to an accept if these are addressed (esp. 1 and 3, with a very strong suggestion for 4 and 2).

1. It's really hard to draw conclusions about robustness-accuracy tradeoffs from Tables 2 and 3. The claim that your method performs "close" to SOTA needs some added substantiation. I'd like:

  *  1.A) Standard deviations on at-least three runs for each setting. In many of these datasets, a 1pp improvement is significant. But accuracies on adversarial datasets often have high variance, so it's unclear how to interpret when your method isn't clearly better and often appears to be much below a handful of the other methods.

More details: TMD more reliably seems to perform better esp, on Imdb:Roberta, RanMASK and SAFER sometimes outperform on  robustness to attack/clean accuracy). It's hard to verify what "close" is, because there are no standard deviations, and on a few of the tasks, AdvFooler really does appear to significantly underperform a couple of the other methods by a  bit. (It's still reliably top 3 among the methods though, and I don't think you need to be better than all to have a case for publication).

   * 1.B. Tune SAFER and RunMASK (tune the rate of substitution/masking) until either the robustness or the accuracy matches your method, so. you can draw a clean comparison. (Or tune your method to their accuracy if that's easier). As is, not clear at all that one is Pareto dominant (or even comparable) to the other.

Example: While RunMASK does suffer on accuracy, it makes things more robust (on AGNEWS dataset BERT for instance). Based on the data available, I'd say it's a toss up whether it's better or worse than AdvFooler.



2. I think the most compelling case for publication is how lightweight your system is. I would include compute time results somewhere. I was unable to find them). It's also unclear to me that simple adversarial training on word substitutions is actually significantly more compute intensive (generating synonyms is super quick, and training on more examples isn't too much of an added burden when you're already finetuning -- esp. if you're comparing to needing to tune your noise param for AdvFooler, which while cheap is still some added compute time).


3. In your system, HotFlip is not perhaps an accurate white-box attack (Appendix C7). HotFlip relies on gradients, but are you giving it the unnoised gradients, or the parameters across all iterations of the model (so someone can reverse engineer the likely original unnoised parameters?). I suspect that if it were truly a white box attack, the certified robustness systems and TMD would outperform. I think these results as presented are a bit misleading otherwise, and one option is to remove them. I would suggest either demonstrating the white-box capabilities fully by (1) giving access to unnoised params, or each set of params so you can derive unnoised params AND (2) comparing to TMD and the certified methods. OR I'd like to see the claim about the defense being "attack agnostic" in the main paper, and the claims of being robust to "white-box attack" toned down.

4. New Baseline Suggestion 1/2: While I understand that it would not be as lightweight as AdvFooler, the style of your method, suggests that a comparison with a diverse ensemble designed to induce adversarial robustness would be very apt since it also tries to induce robustness in much the same way (added random noise) (see my strengths section for the analogy). (I suggest this one: https://proceedings.mlr.press/v97/pang19a , the ADP regularizer)

5. New Baseline Suggestion 2/2: Choose Dirichlet Neighborhood Ensemble (DNE)  over ASCC. While both methods model the perturbation space as the convex hull of word synonyms, DNE far outperforms ASCC (See TMD paper Table 1 where DNE nearly outperforms TMD on AGNews while maintaining a high clean accuracy).

**Questions:**

1. How often are are the predictions for F_advfooler the same as those for F under different noise rates? (Claim in 3.1) The chart showing adversarial robustness dropping while clean accuracy only slightly drops, gets at this question somewhat. But if possible, I'd like something that's even cleaner of a comparison that can maybe help provide intuition for why the clean accuracy doesn't really suffer much (Esp on IMDB) while adversarial robustness totally craters under high noise (Fig 4).

Suggestions for added baselines (2 of them) included in the Weaknesses section (numbers 4 and 5).

**Details Of Ethics Concerns:**

No ethnical concerns really. It's a standard adversarial robustness defense paper for word substitution attacks.

---

> ### Author Response · Authors · 2023-11-21
>
> **Q1: It's really hard to conclude robustness-accuracy tradeoffs from Tables 2 and 3.... Based on the data available, I'd say it's a toss-up whether it's better or worse than AdvFooler.**
>
>
> **A: ** Thank you for the valuable comment. We have revised the performance discussion in Section 4.2, and updated the submission. Indeed, AdvFooler is reliably among the top-3 defenses. However, as mentioned in Section 4.2's discussion, AdvFooler possesses other advantages, such as significantly lower training and inference overhead, being plug-and-play, and requiring no access to the training data.
>
> In our experimments, the calculated the p-values of AdvFooler and the better defenses are below 0.01, indicating that the performance results are statistically significance. We also added this into Supplementary of the revised submission.
>
> We really appreciate the interesting suggestion to "tune SAFER and RunMASK until either the robustness or the accuracy matches AdvFooler". We provide the results when tuning RandMASK to a similar accuracy of AdvFooler. As we can observe, the accuracy under attack of RandMASK decreases tremendously, reducing to half of the original AuA. This shows that, while RanMASK was able to be robust against adversarial attacks, this method also requires larger trade-offs in accuracy compared to other defenses. We added this experiment into Section C.8 (Supplementary), including the experimental setup.
>
> |                              |        | TextBugger AuA(%) (ASR(%)↓) | TextBugger Avg. Query↓ | TextFooler AuA(%) (ASR(%)↓) | TextFooler Avg. Query↓ | BERTAttack AuA(%) (ASR(%)↓) |  BERTAttack Avg. Query↓ |
>  |------------------------------|--------|:---------------------------:|:----------------------:|:---------------------------:|:----------------------:|:---------------------------:|:-----------------------:|
>  |   BERT_MASK (0.9 mask rate)  | 90.14% |       52.50% (41.60%)       |         582.21         |       54.60% (38.93%)       |         511.98         |       61.10% (31.73%)       |          595.57         |
>  |   BERT_MASK (0.3 mask rate)  |  93.5% |       22.50% (75.88%)       |         360.12         |       24.10% (74.09%)       |         325.63         |       42.90% (54.12%)       |          412.99         |
>  | ROBERTA_MASK (0.9 mask rate) | 90.41% |       52.30% (41.50%)       |         575.94         |       54.20% (39.37%)       |         516.98         |       60.00% (32.43%)       |          594.93         |
>  | ROBERTA_MASK (0.3 mask rate) |  93.3% |       34.00% (63.56%)       |         396.72         |        35.7% (61.74%)       |         359.52         |       45.60% (50.97%)       |          429.72         |
>
>
> **Q2:I think the most compelling case for publication is how lightweight your system is...if you're comparing to needing to tune your noise param for AdvFooler, which while cheap is still some added compute time).**
>
> **A:** In Table 5, Section C.1, we provided the inference time of AdvFooler  the AdvFooler and other defenses; AdvFooler only adds a negligible inference overhead to the model, while TMD/RandMASK, SAFER, and ASCC all have non-trivial overhead.
>
> Let N_a be the number of noise scales in AdvFooler. Then, for each noise scale, AdvFooler needs to forward through a small test set. Below, we show the training time for FreeLB and InfoBERT and base-training time + tuning time for AdvFooler (for fair comparison), on AGNEWS. We can observe that, FreeLB and InfoBERT incur significant training overhead, compared to the tuning+base-model training time of AdvFooler.
>
>
> | Model     | AGNEWS                   (s) |
> |-----------|------------------------------|
> | BERT-base |                        12150 |
> | FREELB    |                        52370 |
> | INFOBERT  |                        76650 |
> | AdvFooler |                 12150+40*N_a |

---

> > ### Author Response · Authors · 2023-11-21
> > **Continue the rebuttal comments.**
> >
> > **Q3: In your system, HotFlip is not perhaps an accurate white-box attack (Appendix C7)... I believe that the second scenario might be more realistic and will give a better result compare to the first scenario)**
> >
> > Thank you for the suggestion. Following your suggestion, we disable the noise in AdvFooler when calculating the gradient, letting HotFlip create the adversarial examples using the base model. Similarly, for RandMASK and TMD, we give the adversary access to the base model.
> >
> >
> >  | Model     | AGNEWs [AUA%(ASR%)] | IMDB [AUA%(ASR%)] |
> >  |-----------|---------------------|-------------------|
> >  | BERT-Base | 37.7% (59.98%)      | 59.3 (35.82%)     |
> >  | FREELB    | 51.1% (45.87%)      | 62.3% (31.08%)    |
> >  | INFOBERT  | 43.3% (54.08%)      | 50.3% (44.66%)    |
> >  | AdvFooler | 48.3% (48.50%)      | 77.3% (16.61%)    |
> >  | RanMASK   | 85.3% (3.9%)        | 84.1% (9.8%)      |
> >  | TMD       | 44.2% (52.72%)      | 63.3% (29.9%)     |
> >
> > The results show that AdvFooler has better robustness than TMD, slightly lower than FreeLB on AGNEWS but better on IMDB. RandMASK still has the best robustness, but as indicated, its clean accuracy drops by more than 4%. Given the results, we also think that rigorous evaluation of the existing black-box defenses against white-box attacks can be an interesting, independent work.
> >
> >
> > **Q4: New Baseline Suggestion 1/2: suggests that a comparison with a diverse ensemble designed to induce adversarial robustness**
> >
> > Thank you for your suggestion. Although AdvFooler with different noise vectors can be treated as individual members in the diverse ensemble, our method achieve adversarial robustness from a different principle. For ADP, the diversity in non-maximal predictions reduces the transferability of adversarial samples across the members of the ensemble. On the other hand, we only apply a single noise for each query but do not ensemble the predictions, resulting in a randomized model. As discussed in Theorem 1 of Section 3.2, this randomizing property help the model mislead the attack and thus make it more challenging for the adversary to find adversarial samples. We will include this discussion in the revision.
> >
> >
> > **Q5: Choose Dirichlet Neighborhood Ensemble (DNE) over ASCC.**
> >
> > **A:** As suggested, we trained the Dirichlet Neighborhood Ensemble using the TextDefender implementation for both dataset, with both BERT and RoBERTa as the pre-trained models.
> >
> >
> > |                          |        | TextBugger AuA(%) (ASR(%)↓) | TextBugger Avg. Query↓ | TextFooler AuA(%) (ASR(%)↓) | TextFooler Avg. Query↓ | BERTAttack AuA(%) (ASR(%)↓) |  BERTAttack Avg. Query↓ |
> > |--------------------------|--------|:---------------------------:|:----------------------:|:---------------------------:|:----------------------:|:---------------------------:|:-----------------------:|
> > |      BERT_DNE_AGNEWs     | 94.21% |       35.20% (62.15%)       |         335.39         |       37.70% (59.42%)       |         305.36         |       54.80% (41.01%)       |          360.91         |
> > | BERT_AdvFooler_AGNEWs    | 93.67% |       50.10% (45.90%)       |         819.44         |       50.10% (45.90%)       |         701.05         |       53.40% (42.89%)       |          752.61         |
> > |    ROBERTA_DNE_AGNEWs    | 94.89% |       51.80% (45.19%)       |         425.38         |       52.60% (44.34%)       |         378.81         |       57.60% (39.05%)       |          413.23         |
> > | ROBERTA_AdvFooler_AGNEWs | 94.21% |       51.70% (44.88%)       |         804.99         |       53.30% (42.81%)       |         717.31         |       57.80% (38.38%)       |          767.47         |
> > |       BERT_DNE_IMDB      | 90.15% |       29.40% (67.62%)       |         1470.86        |       30.10% (67.46%)       |         1333.54        | 35.70% (61.28%)             | 1674.86                 |
> > | BERT_AdvFooler_IMDB      |  91.9% |       42.40% (53.41%)       |         3261.41        |       49.10% (47.32%)       |         2759.37        |       40.70% (55.76%)       |         2645.36         |
> > |     ROBERTA_DNE_IMDB     | 95.54% |       56.30% (40.96%)       |         1321.3         |       59.30% (37.97%)       |         1292.96        |       46.90% (50.48%)       |          949.42         |
> > | ROBERTA_AdvFooler_IMDB   | 92.69% |       60.30% (35.58%)       |         4349.38        |       64.70% (30.20%)       |         3360.81        |       50.20% (45.55%)       |         3274.13         |
> >
> >
> > DNE’s accuracy under attack in most case are quite high, being able to surpass ASCC, FreeLB, and InfoBERT in most case, as shown in DNE’s performance on IMDB and AGNEWs dataset. However, AdvFooler still outperforms DNE except in one or two scenarios. This shows that AdvFooler is consistently competitive or sometimes better than adversarial training and ensemble defenses.

---

> > > ### Comment · Reviewer_2d4f · 2023-11-21
> > > **Lingering issues on Q3**
> > >
> > > I strongly appreciate the thoroughness of the rebuttal but I remain unconvinced of thr point that advfooler is even performant on white box attacks.
> > >
> > > Generating attacks on the base model in a white box fashion and seeing whether those attacks transfer to AdvFooler after adding noise is still not a white box attack. The attacker needs to be aware and make good use of the specific noise values added (or even the distribution of noises if using some iterative bounding protocol type attack)
> > >
> > > Without this added information, the attack you describe is not a white box attack for this setting. It's some sort of attack transferability assessment.

---

> > > ### Comment · Reviewer_2d4f · 2023-11-21
> > > **Point taken on Q4. Analog would actually be selecting a model at random from an ensemble.**
> > >
> > > I'm not sure this experiment is needed for publication, but point taken.
> > >
> > > The equivalent would be running adp, and then selecting one model within that system at random per query.
> > >
> > > No further action needed here.

---

> > ### Comment · Reviewer_2d4f · 2023-11-21
> > **Make it more clear that being sota is not a contribution, being in the same ballpark is**
> >
> > I very much appreciate your thoroughness and believe there is strong value to this method even if it doesn't beat SOTA.
> >
> > Can you, however, make it even more clear in the contributions that while you achieve close to sota under more constraints, the contribution is not necessarily to achieve SOTA. I think this would also help frame rhe readers perspective early on.

---

> ### Comment · Reviewer_2d4f · 2023-11-21
> **This is the main remaining issue for me.**
>
> I still think the claim of being robust to white box attack is misleading, and this issue is still blocking me from giving a higher rating.
>
> I suggest you either frame it as an attack transferability from the base model type of issue instead of mentioning white box attack. Or run a white box attack aware or the noise values added.

---

> ### Author Response · Authors · 2023-11-22
> **Our responses to the reviewer's new comments!**
>
> **Q3.1: Generating attacks on the base model in a white box fashion and seeing whether those attacks transfer to AdvFooler after adding noise is still not a white box attack. The attacker needs to be aware and make good use of the specific noise values added (or even the distribution of noises if using some iterative bounding protocol type attack)**
>
> **Without this added information, the attack you describe is not a white box attack for this setting. It's some sort of attack transferability assessment.**
>
> This is again a very useful and insightful suggestion. We updated the paper (Section C.7 in the revised submission) to indicate that these are *not truly white-box attacks*.
>
> We dubbed this attack "transferred white-box" attacks, due to the fact that the adversary has access to the base model (or even the randomized model), but there still exists a mismatch between the attacker's possessed model and the defender's possessed model due to the added randomization (or the attacker does not make good use of the noise distribution, which is not a designed in the existing white-box approach used for the study), thanks to the reviewer's suggestion. We also believe that developing a true white-box attack that makes good use of the noise values is an interesting and independent extension of our work.
>
> **Q6: I very much appreciate your thoroughness and believe there is strong value to this method even if it doesn't beat SOTA. Can you, however, make it even more clear in the contributions that while you achieve close to sota under more constraints, the contribution is not necessarily to achieve SOTA. I think this would also help frame rhe readers perspective early on.**
>
> Thank you for the comment. As suggested, we have revised our paper with the following statement in contributions: "The results demonstrate that AdvFooler is a competitive defense, compared to the existing representative textual adversarial defenses, while being under more constraints, including few modeling assumptions, being pluggable, and incurring negligible additional computational overhead."
>
> **Q4.1: I'm not sure this experiment is needed for publication, but point taken.
> The equivalent would be running adp, and then selecting one model within that system at random per query. No further action needed here.**
>
> Thank you for the comment.
>
> **Q7: I still think the claim of being robust to white box attack is misleading, and this issue is still blocking me from giving a higher rating.
> I suggest you either frame it as an attack transferability from the base model type of issue instead of mentioning white box attack. Or run a white box attack aware or the noise values added.**
>
> We have revised the paper to clarify that this is not truly a white-box attack, in our response to Q6. Again, we appreciate the comment which has helped us improve the clarity of the paper on white-box attacks.
>
> ---
>
> Thank you the reviewer for the insightful comments. We hope that your concerns and comments are now addressed. Nevertheless, we're happy to discuss any additional comments you may have!

---

### Official Review · Reviewer_bLP9 · 2023-10-21

**Soundness:** 2 fair
**Presentation:** 4 excellent
**Contribution:** 2 fair
**Rating:** 3
**Confidence:** 5

**Summary:**

In this work, the authors propose a simple defense method AdvFooler against textual adversarial attacks. Specifically, AdvFooler adds a Gaussian noise at each layer for forward propagation to randomize the latent defense. With such randomization, the attacker cannot find significant words for substitution, making the model robust.

**Strengths:**

1. The paper is well-written and easy to follow.

2. AdvFooler is simple and seems to be effective against several attacks.

**Weaknesses:**

1. The motivation is not clear. Why does such randomization fool the attackers while not degrading the benign performance?

2. Why does AdvFooler can only perplex query-based black-box attacks? It is significant for a defense method to defend against various attacks, such as white-box attacks [1], decision-based attacks [2,3], and so on. It is necessary to validate the effectiveness against these attacks to show the generality of AdvFooler.

3. AdvFooler does not outperform the SOTA baselines against various attacks.

4. From Figure 4, it might be hard to choose a consistent noise scale for different datasets and models.

[1] Wang et al. Adversarial Training with Fast Gradient Projection Method against Synonym Substitution based Text Attacks. AAAI 2021.

[2] Maheshwary et al. Generating Natural Language Attacks in a Hard Label Black Box Setting. AAAI 2021.

[3] Yu et al. TextHacker: Learning based Hybrid Local Search Algorithm for Text Hard-label Adversarial Attack. EMNLP 2022.

**Questions:**

See weakness

---

> ### Author Response · Authors · 2023-11-21
> **Thank you for your invaluable comments.**
>
> **Q1: The motivation is not clear. Why does such randomization fool the attackers while not degrading the benign performance?***
>
> **A:** Thank you for the comment. As explained in Section 3.2 and Theorem 1, the proposed latent feature randomization can fool the query-based attackers since it can mislead the search direction of the adversary's querying process. However, this randomization *will affect/degrade the benign performance*, a fact we specifically mention in Section 3.3. AdvFooler and other randomization-based defenses, such as RandMask and SAFER, will affect the benign performance. As demonstrated in Figure 4, increasing the noise scale leads to a gradual decrease in benign accuracy; at some point, the large noise scale can even lead to a significant decrease in benign performance. We suggest, in Section 4.3 discussing this figure, that the defender should choose a noise scale for their need (e.g., noise scale corresponds to 1% drop in benign accuracy).
>
>
> To summarize, we'd like to clarify that AdvFooler's randomization can fool the attackers but will induce a small drop the benign performance. In addition, unlike RandMASK and SAFER, it only has negligible training and inference overhead and can be plugged into any already trained model for better robustness (Table 1).
>
> **Q2: Why does AdvFooler can only perplex query-based black-box attacks? It is significant for a defense method to defend against various attacks, such as white-box attacks [1], decision-based attacks [2,3], and so on. It is necessary to validate the effectiveness against these attacks to show the generality of AdvFooler.**
>
> **A:** We'd like to emphasize the practicality of threat model discussed in the paper. For many companies with ML applications, the trained model as well as training data are valuable assets to the companies, thus are secretly guarded. Thus, the attacker rarely has access to the trained model or model’s architecture to perform white-box attacks. Instead, the attacker will rely on querying the models so that they can estimate the optimal perturbation toward adversarial examples, as discussed in Section 1. The same black-box, query-based threat model is frequently examined in several related works [(Nguyen Minh and Luu, 2022), (Zeng et al, 2021), (Dong et al, 2021), (Si et al, 2021), (Li et al, 2021)].
>
> Nevertheless, we also agree that, for broad applicability, it is necessary to validate the effectiveness of AdvFooler against white-box and decision-based attacks, which we already provided in C.7 and C.6 (Supplementary), respectively. These experiments confirm that AdvFooler is still effective against the evaluated white-box and decision-based attacks. As discussed in these experiments, we conjecture that, for white-box attacks, AdvFooler's randomization nullifies the adversarial perturbation and brings it back to the correct side of the decision boundary; since decision-based attacks still involve querying the model, AdvFooler's randomization can return incorrect prediction for some perturbed samples, especially near the decision boundary, and therefore misleads the attacks' optimization process.

---

### Official Review · Reviewer_9Kvp · 2023-11-01

**Soundness:** 3 good
**Presentation:** 3 good
**Contribution:** 2 fair
**Rating:** 6
**Confidence:** 3

**Summary:**

This paper proposes a lightweight and attack-agnostic defense method to against query-based black-box attacks called AdvFooler. AdvFooler accomplishes this by introducing randomization to the latent input representation during inference.  The advantage of Advfooler is that it does not need additional computational overhead during training nor relies on assumptions about the potential adversarial perturbation set.

**Strengths:**

1. Compared with other defense methods, the proposed method AdvFooler is simple, pluggable and does not require additional computational overhead during testing or access to training.
2. The authors conducted comprehensive experiments to assess the effectiveness of AdvFooler, employing two BERT models, two distinct datasets, and three different attack methods. Furthermore, they provided qualitative analyses of their results.

**Weaknesses:**

1. Despite its advantages in terms of simple implementation and minimal computational overhead, AdvFooler's performance falls short of the state-of-the-art. In Table 2, on the AGNEWS dataset, AdvFooler exhibits lower accuracy under attack compared to RanMASK for the BERT-base model, and it also demonstrates lower accuracy under attack than both TMD and RanMASK for the RoBERTa-base model.

2. The selection of the hyper-parameter for noise scale in AdvFooler is not entirely clear. The authors claim that they choose the ν value based on the criterion that the clean accuracy drops by at most 1% using the test set. However, it seems not the case in Table 2 and 3. In Figure 4, the curves of AuA are not monotone. Do authors also consider AuA values when choosing noise scale?

**Questions:**

Instead of adding noises to all layers by the same noise scale, what would the results be if adding different noise scale to different layers?

---

> ### Author Response · Authors · 2023-11-21
> **Thank you for the valuable comments!**
>
> Please see our responses below.
>
> **Q1: Despite its advantages in terms of simple implementation and minimal computational overhead, AdvFooler's performance falls short of the state-of-the-art. In Table 2, on the AGNEWS dataset, AdvFooler exhibits lower accuracy under attack compared to RanMASK for the BERT-base model, and it also demonstrates lower accuracy under attack than both TMD and RanMASK for the RoBERTa-base model.**
>
> We acknowledged, in Section 4.2, that AdvFooler has slightly lower performance than RandMASK and TMD in some experiments.
>
> But, as explained in Section 1, AdvFooler has other important advantages compared to RandMASK and TMD: (1) having negligible computation overhead while (2) simultaneously achieving comparable robustness to RandMASK and TMD, and (3) being plug-and-play, which allows it to be used without modifying existing parameters of the model. Simultenously achieving (1) and (2) is an extremely challenging task; for example, TMD and RandMask require significant additional training and inference overhead for similar robustness to AdvFooler. Also for, for RandMASK, its clean accuracy drops significantly (>4% for RandMASK vs. < 1% for AdvFooler). For these reasons, AdvFooler is significantly more practical for usage in real-world applications, a significant contribution to the adversarial defense domain.
>
> **Q2: The selection of the hyper-parameter for noise scale in AdvFooler is not entirely clear. The authors claim that they choose the $\nu$ value based on the criterion that the clean accuracy drops by at most 1% using the test set. However, it seems not the case in Table 2 and 3. In Figure 4, the curves of AuA are not monotone. Do authors also consider AuA values when choosing noise scale?**
>
> **A:** We'd like to, first, re-iterate the discussion in Section 3.3: "the defender can select the noise scale $\nu$ at which the variance causes the clean accuracy drops by a chosen percentage (e.g., in our experiments, it is 1\%)". In practice (as discussed in threat model), the defender only has the trained model and a small clean test set and makes no assumption about the attacks; using the clean test set, defender finds the largest noise scale corresponding to a certain, tolerable drop in clean accuracy. For example, given the values we used to plot Figure 4 as in the table below, starting with the original clean accuracy of 95.14%, the defender would choose $\nu=0.6$, which corresponds to 94.57% clean accuracy, assuming a selection of 1% drop in performance. For 2% drop (or 94.57% clean accuracy), $\nu$ is set to 0.80. As we can see, the larger $\nu$ generally increases the robustness (43.90% and 48.80% AuA for $\nu=0.6$ and $\nu=0.8$, respectively, on TextFooler); however, when $\nu$ is too large, as we can observe in this figure, it will also interfere clean accuracy (a sudden, significant drop in clean accuracy) and correspondingly decrease the robustness of the model as well.
>
>
>
> | noise_type\intensity | 0.10   | 0.20   | 0.30   | 0.40   | 0.50   | 0.60   | 0.70   | 0.80   | 0.90   | 1      | 1.10   | 1.20   | 1.30   | 1.40   | 1.50   |
> |:--------------------:|--------|--------|--------|--------|--------|--------|--------|--------|--------|--------|--------|--------|--------|--------|--------|
> | Clean (95.14%)       | 95.17% | 95.06% | 95.04% | 94.94% | 94.69% | 94.57% | 94.11% | 93.67% | 93.12% | 92.48% | 90.74% | 87.79% | 83.18% | 77.04% | 70.05% |
> | AuA (TextFooler)     | 38.30% | 38.40% | 41.10% | 43.90% | 45.60% | 48.10% | 48.80% | 50.10% | 45.50% | 35.50% | 21.90% | 13.30% | 8.40%  | 5.80%  | 4.40%  |
> | AuA (TextBugger)     | 36.00% | 37.10% | 38.20% | 40.40% | 45.60% | 46.90% | 49.90% | 50.10% | 44.40% | 33.40% | 20.20% | 13.70% | 8.20%  | 5.40%  | 4.20%  |
> | AuA (BERTAttack)     | 46.70% | 48.80% | 50.30% | 47.50% | 50.30% | 55.40% | 54.50% | 53.40% | 48.60% | 36.20% | 22.10% | 12.10% | 7.10%  | 5.10%  | 4.40%  |
>
>
>
> **Q3: **Instead of adding noises to all layers by the same noise scale, what would the results be if adding different noise scales to different layers?****
>
> **A:** Thank you for the interesting question. We provide the experiments (AGNEWs) for AdvFooler with different noise scales added to different layers (we multiply the original noise scale by the standard deviation of that layer’s output)
>
> | Model     | TextFooler [AUA%(ASR%)] | TextBugger [AUA%(ASR%)] | BERTAttack       [AUA%(ASR%)] |
> |-----------|-------------------------|-------------------------|-------------------------|
> | AdvFooler |          52.0% (44.21%) |         49.5% (47.96%)  | 55.9% (39.83%)          |
>
> The results show that this variation has good performance compared to our current AdvFooler’s version.

---

### Official Review · Reviewer_UMzy · 2023-11-07

**Soundness:** 3 good
**Presentation:** 3 good
**Contribution:** 2 fair
**Rating:** 5
**Confidence:** 5

**Summary:**

The authors observe that adding noise to hidden layers can result in obscuring identification of important words in a sentence, a crucial step needed for launching attacks. They evaluate their proposal and highlight efficacy.

**Strengths:**

1. Low cost defense.

**Weaknesses:**

1. Does not perform better than prior works.
2. Similar in ideology to DP.

**Questions:**

1. Could the authors compare and contrast their approach to applying DP-style noise to the embeddings generated by the hidden layer?
2. Results from Table 2 suggest that their approach is not the very best compared to other approaches. What are the merits of the proposal then? Apart from reduced computational overheads?
3. The authors empirically note that adding small amounts of noise to the logins does not change the prediction. But this is not fundamental nor clear why this is the case. Could the authors elaborate?
4. Why can’t the adversary utilize a proxy model to launch its attacks instead of the current black-box setup i.e., use the proxy model and attention values to identify important words?
5. Alternatively, the adversary could also replace subsets of words using brute force. For small spans of text (as considered in the evaluation), this is not computationally prohibitive. Could the authors elaborate further the specific scenarios where important word selection inhibition is prudent?

---

> ### Author Response · Authors · 2023-11-21
> **Thank you for the valuable comments!**
>
> Please see our responses to your comments below:
>
> **Q1: **Could the authors compare and contrast their approach to applying DP-style noise to the embeddings generated by the hidden layer?****
>
> **A:** Thank you for the comment; however, we’re not sure what DP stands for and will assume DP refers to Differential Privacy, a privacy protection framework that introduces a small change in the data for a guaranteed, bounded change in the distribution of the algorithm’s outputs. DP-style defenses such as (Lécuyer et al 2019) certify the defense under the DP framework; it is important to note that SAFER and RandMASK, evaluated in the paper, are also certified defenses and share a similar approach to DP-style defenses. DP-defense and RandMASK still require training overhead to tune the added noise so that it does not interfere with the model's prediction. On the other hand, AdvFooler adds noise to the latent features at inference time to misdirect the attacker's querying process.
>
> We have added this discussion to our revised submission.
>
>
> Lécuyer, M., Atlidakis, V., Geambasu, R., Hsu, D.J., & Jana, S.S. (2018). Certified Robustness to Adversarial Examples with Differential Privacy. *2019 IEEE Symposium on Security and Privacy (SP)*, 656-672.
>
>
> **Q2: **Results from Table 2 suggest that their approach is not the very best compared to other approaches. What are the merits of the proposal then? Apart from reduced computational overheads?****
>
> **A:** As explained in Section 1 (especially Table 1), AdvFooler (1) reduces significant computation overhead while (2) simultaneously achieving comparable robustness to other defenses. This is an extremely challenging task; for example, TMD/RandMask/SAFER or adversarial training approaches require significant additional training overhead for similar robustness to AdvFooler; TMD/RandMask/SAFER also incurs non-trivial inference overhead, making them less practical in real-world application. Another advantage of AdvFooler is (3) plug-and-play, as also indicated in Table 1, which allows it to be used without modifying existing parameters of the model. Finally, as indicated in Section 3.3 and 4, AdvFooler allows the user to (4) control how much accuracy they want to trade-off for robustness using a small test set. (1), (2), (3) and (4) make AdvFooler significantly more practical to employ in real-world applications, a significant contribution to the adversarial defense domain.
>
> **Q3: The authors empirically note that adding small amounts of noise to the logins does not change the prediction. But this is not fundamental nor clear why this is the case. Could the authors elaborate?**
>
> **A:** Defenses via adding noise (including AdvFooler and RandMASK/SAFER) will inevitably affect the prediction or clean accuracy. While RankMASK/SAFER perturbs the input and require additional training and/or inference overhead to tune the added noise for a small accuracy drop, AdvFooler adds noise to the hidden features only at inference time, an approach which is theoretically shown to be able to fool the attacker (discussed in Section 3.2). To choose this noise, however, we suggest using a small test set and finding a large-enough noise for a specific tolerance of accuracy drop (discussed in Section 3.3 and Figure 4). This allows the defender to control how much accuracy they want to trade off and experiments show that our approach can achieve comparable robustness with other defenses while having only <1% accuracy drop.

---

> > ### Author Response · Authors · 2023-11-21
> > **Continued the rebuttal comments.**
> >
> > **Q4: Why can’t the adversary utilize a proxy model to launch its attacks instead of the current black-box setup i.e., use the proxy model and attention values to identify important words?**
> >
> > **A:** As mentioned in the Limitation section, our work focuses on the black-box, query-based attacks, similar to the related works [(Nguyen Minh and Luu, 2022), (Zeng et al, 2021), (Dong et al, 2021), (Si et al, 2021), (Li et al, 2021)]. Since AdvFooler is designed to mislead the *querying process* and black-box, proxy-based attacks (i.e., those that rely on third-party proxy language models) do not have a querying process,  studying AdvFooler’s effectiveness against these attacks is an interesting extension of our work.
> >
> > On the other hand, HotFlip, evaluated in Section C.7 (Table 12), is an attack that has *access to the true model*, which makes it a stronger attack than a black-box attack with a proxy model. AdvFooler can still be effective against HotFlip. For this reason, we believe that AdvFooler will still be effective against black-box attacks with proxy-models.
> >
> >
> > **Q5: Alternatively, the adversary could also replace subsets of words using brute force. For small spans of text (as considered in the evaluation), this is not computationally prohibitive. Could the authors elaborate further on the specific scenarios where important word selection inhibition is prudent?**
> >
> > **A:** Thank you for the interesting suggestion. As discussed Section 1, adversarial black-box query-based attack methods manipulate different aspects of the input sentence, such as replacing words, without significantly changing the original semantic in the perturbed inputs while limiting the number of queries to the model within *a budget*. In a realistic and standard evaluation setting (considered in our paper, Section 4, and others in the literature), an adversary is only allowed to query at most $50 * L$ (L: number of words), otherwise, the attack is *practically* infeasible. For a text with 38 words (average length of AGNEWS), assuming just 3 synonyms (on average) for each word, the number of queries should be 3^38 >> 50 * 38; when it's the 90th percentile, it's still 3^27 >> 50 * 38. For these reasons, an effective adversarial text attack must consider a better strategy than brute-force solutions.
> >
> > For brute-force attacks to be realistic, the text length should be quite small. In this scenario, we believe that AdvFooler, as well as other defenses, will not be effective if at least one of the perturbed versions of the text are an adversarial example.
> >
> > In Section C.6 (Table 11), on the other hand, we evaluated AdvFooler against a more-efficient brute-force style attack, called Hard-label, where the attack first randomly perturb (without consideration for preserving the semantic of a perturbed sample) the input until a perturbed sample flips the prediction. This is a more efficient, brute-force process since not paying attention to semantic preservation means the attack can find the perturbed sample with a different prediction very quickly. Then, in the second step, the attack moves this sample closer to the original sample until they're semantically similar. Still, AdvFooler is effective against Hard-label attacks.

---

### Author Response · Authors · 2023-11-22
**Thank you for your valuable comments and welcome additional questions!**

We want to start by thanking the reviewers once more for providing constructive feedback and raising questions to help us improve the quality of our paper. We have addressed all the questions and suggestions from the reviewers in our rebuttal with new experiments and paper revisions (please see our revised submission) as recommended. Please share your additional thoughts and we're happy to address them promptly.

---

### Author Response · Authors · 2023-11-23
**Thank you again for helping improve our work!**

Our work proposes AdvFooler, a lightweight defensive method that improves the robustness of a model against textual adversarial attacks. AdvFooler achieves competitive robustness compared to the state-of-the-art, consistently being in top-3 robust defenses, while being under more constraints, including few modeling assumptions, no access to training, being pluggable, and incurring negligible computational overhead during inference.

During the rebuttal process, we have:

* Provided additional experiment details, including rationale on noise scale selections.
* Provided additional analysis of why AdvFooler works, especially on the analysis of noise's effect on clean accuracy and AuA.
* Provided additional experiments on other defense methods, such as DNE and RanMASK under different noise scales.
* Provided additional experiments on different attack methods
* Clarified that the originally-named white-box attack is in fact white-box attacks based on transferrability.
* Provided additional comparisons with other adversarial defense methods on training cost.
* Provided references to sections in our paper to answer the reviewers' questions.

We have added these discussions in the revised submissions. We hope that our responses have addressed the concerns of the reviewers. AdvFooler is a highly applicable defense method with many additional advantages and competitive performances. Finally, we're happy to promptly answer additional questions.

---

### Meta-Review · Area_Chair_1HuJ · 2023-12-08

**Metareview:**

This paper proposes AdvFooler, which is a defense method against single word text attacks. The proposed defense is based on perturbing the latent representation of the input to alter the importance of the words to the extent that another word appears to be the most important. The authors perform several experiments to showcase the effectiveness of their method, which is consistently in the top 3 defense strategies across different model architectures, making it an effective defense against the tested attacks. Here is my concerns against the publication of the paper.

1. The paper's writing is confusing. I think the paper should make it clear that the focus is on single-word substitution attacks from the beginning; I only realized this when the attack was formulated in Section 3.

2. The choice of hyperparameters seems to be arbitrary for achieving the clean accuracy. Ideally, each attack should produce a tradeoff curve between clean accuracy and adversarial accuracy (or other adversarial metrics), and then an attack would be more successful if the curve dominates another attack's. As it stands some of the comparisons are apple-to-oranges.

3. I think the defense should be evaluated against other attacks, say two word substitutions, paraphrases, etc, to see how well it generalizes. I agree with the reviewers that understanding this aspect is quite important.

4. I think the proposal by Reviewer UMzy in their review is quite a realistic attack against this defense, and it likely breaks this defense: This paper's adversary’s approach relies on identifying important words (tokens); it is claimed that by ensuring that the “wrong” words are identified, attacks will fail. The attacker could use a “proxy” model which it has complete access to, to find important words. The premise of security research is not to “constrain” the attacker, but to assume what it can realistically do.

5. The math is quite sloppy. Theorem 1 is indeed invalid and its proof attempt is incorrect.

Overall, I think this paper has quite a bit of potential and I totally agree with all of the strengths mentioned by Reviewer 2d4f. However, I think the paper needs to be fleshed out a bit more to address the issues raised. While I think having theory to support the proposal would be quite appealing, the paper could still be publishable as a pure empirical study IMHO. I also think that novelty should not be a blocker for publication of this paper as I have not seen the same inference-time defense before, which seems like a nice adaptation of similar perturbation-based defenses such as DP. Finally, my main takeaway from this paper is that single-word substitutions are quite simple to defend against at inference time and that may be an artifact of the attack and not a feature of the defense. As such, my recommendation to the authors would be to:

- Try the attack proposed by Reviewer UMzy and also include more versatile attacks that don't necessarily rely on single-word substitutions.

- Investigate the tradeoffs between clean accuracy and adversarial robustness in a systematic manner, and also try to develop a recipe for hyperparameter selection for the noise for this method.

- Drop the sloppy math, and revise the paper accordingly.

**Justification For Why Not Higher Score:**

Two major reasons that led to reject recommendation

- The empirical substantiation is insufficient, especially it is important to understand the generalization beyond single-word substitution attacks.

- The math is sloppy, and the paper would be better off removing the theoretical claims.

**Justification For Why Not Lower Score:**

The method is a neat and clever defense against single-word substitution attacks.

---

### Decision · Program_Chairs · 2024-01-16

Reject